# Dynamics of Myosin II Filaments during Wound Repair in Dividing Cells

**DOI:** 10.3390/cells10051229

**Published:** 2021-05-17

**Authors:** Md. Istiaq Obaidi Tanvir, Go Itoh, Hiroyuki Adachi, Shigehiko Yumura

**Affiliations:** 1Graduate School of Sciences and Technology for Innovation, Yamaguchi University, Yamaguchi 753-8511, Japan; tanviristiaq@gmail.com; 2Department of Molecular Medicine and Biochemistry, Akita University Graduate School of Medicine, Akita 010-8543, Japan; goitoh@med.akita-u.ac.jp; 3Department of Biotechnology, Graduate School of Agricultural and Life Sciences, The University of Tokyo, Bunkyo-ku, Tokyo 113-8657, Japan; adachih@mail.ecc.u-tokyo.ac.jp; 4Collaborative Research Institute for Innovative Microbiology, The University of Tokyo (CRIIM, UTokyo) Yayoi 1-1-1, Bunkyo-ku, Tokyo 113-8657, Japan

**Keywords:** actin, cell membrane, cleavage furrow, laserporation, myosin, wound repair

## Abstract

Wound repair of cell membranes is essential for cell survival. Myosin II contributes to wound pore closure by interacting with actin filaments in larger cells; however, its role in smaller cells is unclear. In this study, we observed wound repair in dividing cells for the first time. The cell membrane in the cleavage furrow, where myosin II localized, was wounded by laserporation. Upon wounding, actin transiently accumulated, and myosin II transiently disappeared from the wound site. Ca^2+^ influx from the external medium triggered both actin and myosin II dynamics. Inhibition of calmodulin reduced both actin and myosin II dynamics. The wound closure time in myosin II-null cells was the same as that in wild-type cells, suggesting that myosin II is not essential for wound repair. We also found that disassembly of myosin II filaments by phosphorylation did not contribute to their disappearance, indicating a novel mechanism for myosin II delocalization from the cortex. Furthermore, we observed that several furrow-localizing proteins such as GAPA, PakA, myosin heavy chain kinase C, PTEN, and dynamin disappeared upon wounding. Herein, we discuss the possible mechanisms of myosin dynamics during wound repair.

## 1. Introduction

Cell membranes are frequently injured by physical damage, chemicals, or pathogens. The wound repair of single cells is essential for cell survival. Wounded cell membranes lose their barrier function; this results in an influx of undesirable substances into the cell, as well as the loss of cytoplasm. To prevent further damage, cells need to rapidly seal and remodel the wound area. Defects in cell membrane repair may cause muscular dystrophy, diabetes, vitamin deficiencies, and inflammatory myopathy [1,2,3,4,5,6]. Therefore, similar to DNA repair, wound repair is a physiologically vital phenomenon in living cells.

The molecular mechanism of wound repair has been studied in different organism models, including mammalian cells, amphibian eggs, echinoderm eggs, fruit flies, nematodes, amoebae, budding yeasts, and *Dictyostelium* cells [7,8,9,10,11,12,13]. A common feature among them is that Ca^2+^ in the external medium is essential for wound repair. The entry of Ca^2+^ is considered to mediate the closure of wound pores [8,14]. However, the complete molecular mechanism of wound repair remains elusive.

In large cells such as *Xenopus* eggs and *Drosophila* embryos, an actomyosin ring, similar to the contractile ring in dividing cells, surrounds the wound site, and its constriction facilitates closure of the wound pore [7,15]. However, in smaller cells such as yeast cells, animal cultured cells, and *Dictyostelium* cells, only actin transiently accumulates at the wound site [10,16,17]. Under deficiency of actin polymerization, wound pores do not close in *Dictyostelium* cells [18]. However, the role of myosin II in wound repair remains unclear.

As in other models, annexins, which are membrane scaffold proteins, and/or endosomal sorting complexes required for transport (ESCRTs), are also candidates of the wound repair mechanism; these proteins contribute to the closure of wound pores or cut the wound patch. Although annexins and/or ESCRTs immediately accumulate at the wound site, actin accumulation is not very fast [19,20,21,22,23].

Recently, we developed a new method using laserporation and proposed a two-step model for wound repair [18]. In this model, upon wounding, Ca^2+^ enters through the wound pore and triggers the de novo generation of vesicles and mutual fusion of vesicle–vesicle and vesicle–cell membranes to create an urgent membrane plug. Actin accumulates to complete the plug, depending on Ca^2+^ and calmodulin.

Herein, we show that when the cell membrane of the cleavage furrow in dividing *Dictyostelium* cells was wounded by laserporation, myosin II transiently disappeared from the wound site. In contrast, actin transiently accumulated at the wound site. Ca^2+^ influx from the external medium triggered both actin and myosin II dynamics. Calmodulin was an upstream modulator for both of them. We also examined the mechanism of the disappearance of myosin II filaments and found that their disassembly by the phosphorylation of their heavy chains did not contribute to the disappearance, indicating a novel mechanism for myosin II delocalization from the cortex. Moreover, several proteins that localized at the cleavage furrow also disappeared upon wounding. Finally, we discuss the possible mechanisms of myosin II dynamics during wound repair.

## 2. Materials and Methods

### 2.1. Cell Culture

*Dictyostelium discoideum* (AX2) and mutant cells were cultured at 22 °C in a plastic dish containing HL5 medium (1.3% bacteriological peptone, 0.75% yeast extract, 85.5 mM D-glucose, 3.5 mM Na_2_HPO_4_, and 3.5 mM KH_2_PO_4_, pH 6.3), as previously described [24]. For wound experiments, the cells were suspended in HL5 medium supplemented with 3 mM CaCl_2_.

### 2.2. Plasmids and Mutants

GFP-myosin II [25], GFP-3ALA myosin II [24], GFP-E476K myosin II [25], GFP-Lifeact [18], GFP-MHCKC [26], GFP-PakA [27], GFP-dlpA [28], GFP-clathrin light chain [29], GFP-PTEN (G129E) [30], GFP-calmodulin [18], and Dd-GCaMP6s [31] expression vectors have been previously described. These expression vectors were transformed into cells by electroporation or laserporation, as described previously [32,33]. The transformed cells were selected in HL5 medium containing 10 μg/mL G418 (Wako Pure Chemical Corporation, Osaka, Japan) or 10 μg/mL blasticidin S hydrochloride (Wako Pure Chemical Corporation) in plastic dishes.

We constructed an expression vector for GFP-cortexillin II. *CtxB* genes were amplified from a cDNA library by PCR using the following primer sets with restriction enzyme sites (underscore): 5′-ATGGATCCATGGATTTAAATAAAGAATGGGAAAAAGTTC-3′ (BamH1) and 5′-ATGAGCTCTTATTTTTTAGCAGCAGCTTTTGCTTCTTC-3′ (Sac1). *CtxB* fragments were subcloned between the BamH1 and Sac1 sites lying downstream of the C-terminal GFP site in the pA15GFP expression vector.

Myosin II-null cells (HS1) [34], MHCKC-null cells [26], PakA-null cells [27], dlpA-null cells [28], CHC-null cells [35], cortexillin II-null cells [36], GAPA-null cells [37], and PTEN-null cells [38] have been previously described.

The sources of the plasmids and mutants used in this study are listed in the Appendix A.

### 2.3. Chamber Preparation

The surface of the coverslip of a glass-bottom chamber was coated with carbon by vapor deposition, as previously described [33,39]. The thickness of the coating layer was approximately 20 nm. To make the surface hydrophilic, the surface of the coated coverslip was activated by plasma treatment. The chamber was sterilized with 70% ethanol and dried, if necessary. The cells were placed on the surface of the coated coverslip and mildly compressed with agarose block (2%, dissolved in BSS containing 10 mM NaCl, 10 mM KCl, 3 mM CaCl_2_, and 3 mM MES, pH 6.3, 1 mm thick) to observe the ventral cell surface.

### 2.4. Wounding and Microscopy

Cells expressing GFP proteins were observed under a total internal reflection fluorescence (TIRF) microscope (based on the IX71 microscope, Olympus), as previously described [40]. The cells were wounded with a nanosecond-pulsed laser (FDSS532-Q, CryLas) and the wound diameter was set as 0.5 μm, as previously described [39]. Time-lapse fluorescence images were acquired with 40–100 ms exposure times and at 130–500 ms intervals using a cooled CCD camera (Orca ER, Hamamatsu Photonics). The time courses of the fluorescence intensities within the circle (3 µm in diameter) including the wound site were examined using Image J (http://rsbweb.nih.gov/ij (accessed on 17 May 2021)). The fluorescence intensities were normalized by setting the value before wounding to 1 after subtracting the background.

### 2.5. Measurement of Influx of PI and Ca^2+^

To observe its influx, 0.15 mg/mL propidium iodide (Sigma-Aldrich, Tokyo, Japan) was added to the external medium before wound experiments. To monitor the dynamics of cytosolic Ca^2+^, cells expressing Dd-GCaMP6s were wounded using laserporation [18,39]. These probes were illuminated by an argon laser (488 nm) and monitored using TIRF microscopy.

### 2.6. Ca–EGTA Buffer

To formulate a medium containing the indicated free Ca^2+^, a Ca–EGTA buffer (10 mM KCl, 10 mM NaCl, 3 mM MES, 10 mM EGTA, and an appropriate concentration of CaCl_2_, pH 6.3) was used. The concentration of CaCl_2_ in the buffer was calculated using Ca–EGTA Calculator v1.3 (https://somapp.ucdmc.ucdavis.edu/pharmacology/bers/maxchelator/CaEGTA-TS.htm (accessed on 17 May 2021)). The agarose block for the agar overlay was prepared by dissolving 2% agarose in Ca–EGTA buffer, as previously described [41].

### 2.7. Inhibitors

W7 hydrochloride (Funakoshi Co. Ltd., Tokyo, Japan) was dissolved in dimethyl sulfoxide (DMSO) to prepare a 10 mM stock solution. Cells were incubated with a final concentration of 20 μM W7 hydrochloride in the medium for 30 min before wound experiments. Jasplakinolide (Sigma-Aldrich) was dissolved in DMSO to make a 0.5 mM stock solution. The cells were incubated with a final concentration of 8 µM jasplakinolide in the medium for 30 min before wound experiments.

### 2.8. Statistical Analysis

Statistical analysis was performed using GraphPad Prism 7 (GraphPad Software Inc., San Diego, CA, USA). Data were expressed as means ± SD and analyzed using a two-tailed Student’s *t*-test for comparison between two groups, or one-way ANOVA with Tukey’s multiple comparisons test.

## 3. Results

### 3.1. Myosin II Filaments Transiently Disappear from the Wound Site

We used the laserporation method to make a local wound in the cell membrane alone [11,33]. After placing cells on a coverslip coated with carbon by vapor deposition, a laser beam was focused on a small local spot beneath a single cell under a total internal reflection fluorescence (TIRF) microscope. The energy of the laser beam absorbed by the carbon made a small pore in the cell membrane that was attached to the carbon coat (Figure 1A).

Wound experiments have been conducted on interphase cells of various organisms. In this study, we examined wound repair in dividing cells for the first time. Figure 1B shows a typical time course of fluorescence images when a dividing *Dictyostelium* cell expressing GFP-myosin II was wounded at the cell membrane of the cleavage furrow by laserporation. TIRF microscopy showed that individual filaments of myosin II localized at the cleavage furrow, as described previously [42]. To make a wound pore, cells were mildly pressed with an agar block to attach the furrow membrane to the surface of the coated coverslip. Interestingly, upon laser irradiation at the cleavage furrow (wound size: 0.5 µm diameter), myosin II filaments began to disappear 5.8 s (5.8 ± 1.1 s, *n* = 25) after wounding, the dark spot expanded, and they finally reappeared (Supplementary Video S1). Figure 1C shows typical time courses of fluorescence images only at the wound site (GFP-myosin II/furrow). During recovery, myosin II filaments reappeared, not always in a manner closing in from the outside, but evenly in the dark spot. Figure 1D,E show the time course of fluorescence intensities at the wound site (red line in Figure 1D) and a typical time course of the dark spot expansion, recorded by line-scanning across the wound site (Figure 1E), respectively. Figure 1F shows the time course of the expansion of dark spots (area), suggesting that the dark spots expand at an almost constant velocity (0.33 ± 0.17 µm^2^/s, *n* = 19).

We also examined the dynamics of myosin II filaments during wounding at the cell membrane outside the furrow (daughter cortex). Since the density of myosin II filaments was much lower in the daughter cortex, the graph of fluorescence intensities did not change over time (blue line, Figure 1D). However, because multiple myosin II filaments simultaneously disappeared after wounding in the daughter cortex (data not shown), only the initiation time was estimated (4.1 ± 0.6 s, *n* = 25). We also examined the dynamics of myosin II filaments in the cortex of interphase cells. We selected the area with the highest concentration of myosin II to create a wound. The myosin II filaments transiently disappeared in a manner similar to that in the furrow cortex (Figure 1C,G). However, because the interphase cells rapidly migrated away from the wound site after wounding [39], precise measurement was difficult; nevertheless, the initiation time was 3.3 ± 0.6 s (*n* = 25), and this was not affected by cell migration.

These results indicated that the transient disappearance of myosin II after wounding was not specific to the furrow membrane. However, myosin II began to disappear at the cleavage furrow significantly later than at the cortexes of interphase and daughter cells (Figure 1K).

### 3.2. Actin Filaments Transiently Accumulate at the Wound Site

Cells expressing GFP-Lifeact, a marker of actin filaments, were wounded at the furrow membrane by laserporation. Previously, we showed that the time courses of the fluorescence intensities of GFP-Lifeact and GFP-actin were not significantly different [18]. Actin filaments began to accumulate at the wound site 4.1 s (4.1 ± 0.7 s, *n* = 25) after wounding and returned to a resting level approximately 24 s (24.1 ± 3.3 s, *n* = 25) after wounding (Figure 1C,H). Figure 1I shows a typical time course of a line scan of fluorescence intensities across the wound site, indicating that the area of actin accumulation also expanded in a manner similar to the myosin II dynamics.

When the cell membrane outside the furrow (daughter cortex) was wounded, actin filaments transiently accumulated much earlier (2.9 ± 0.7 s, *n* = 25) than at the furrow. Figure 1J shows a comparison of the time courses of the dynamics of actin and myosin filaments, suggesting that the actin dynamics initiated slightly earlier than that of myosin II. Figure 1K shows a summary of the initiation, peak, and termination times of actin and myosin II under the above conditions. Actin filaments also accumulated at the wound site at all locations, independent of cell division, although the initiation was significantly delayed at the furrow membrane.

### 3.3. Both Myosin II and Actin Dynamics Are Regulated by Ca^2+^ and Calmodulin

Previously, we showed that the influx of Ca^2+^ from the external medium is essential for wound repair [11,18]. In the present study, dividing cells expressing Dd-GCaMP6s, a Ca^2+^ sensor, were wounded at the furrow membrane to visualize cytosolic Ca^2+^. Immediately after wounding, the fluorescence from the wound site increased, spread over the cytoplasm, and finally decreased to the resting level (Figure 2A). Figure 2B shows the time course of the relative fluorescence intensities of Dd-GCaMP6s upon wounding in the presence and absence of Ca^2+^ (EGTA), suggesting that Ca^2+^ entered through the wound pore immediately after wounding. 

Next, we examined whether Ca^2+^ influx is required for the actin and myosin II dynamics. Figure 2C,D show the time courses of fluorescence intensities of GFP-Lifeact and GFP-myosin II at a wound site in the cleavage furrow in the presence and absence of Ca^2+^. Neither actin nor myosin II dynamics occurred in the absence of Ca^2+^, suggesting that they depend on the influx of Ca^2+^.

Figure 2E shows the dependency of GFP-myosin II dynamics on the external free Ca^2+^ concentration (the biggest troughs in the fluorescence intensities were plotted), suggesting that free Ca^2+^ concentrations higher than 10^−4^ M in the external medium were required for myosin II dynamics; this is consistent with the requirement for actin accumulation in interphase cells [18].

Previously, we showed that calmodulin contributes to actin accumulation at the wound site [18]. We first confirmed this finding in dividing cells. Figure 2F,G show a typical time course of fluorescence images of GFP-calmodulin at a wound site and the time course of the fluorescence intensities, respectively. GFP-calmodulin accumulated neither in the presence of EGTA (Figure 2G, blue) nor in the presence of W7, an inhibitor of calmodulin (Figure 2G, green), suggesting that calmodulin accumulation depends on Ca^2+^ influx. Figure 2H,I show time courses of fluorescence intensities of GFP-Lifeact and GFP-myosin II at the furrow upon wounding in the presence and absence of W7, suggesting that both actin and myosin II dynamics were significantly inhibited.

These results indicate that both myosin II and actin dynamics are regulated by Ca^2+^ and calmodulin.

### 3.4. Role of Myosin II in Wound Repair

To examine the role of myosin II in wound repair, myosin II-null cells were wounded in a medium containing propidium iodide (PI), which emits fluorescence upon binding to RNA or DNA. Figure 3A shows a typical time course of fluorescence images of PI influx through wound pores. The fluorescence began to increase at the wound site and spread throughout the cytoplasm. Figure 3B shows the time courses of PI influx in wild-type (AX2) and myosin II-null (HS1) cells, indicating that the influx of PI dye ceased approximately 2–4 s after wounding in both cells. Therefore, the wound pore closed within a similar time period, independent of myosin II.

Next, we examined the relationship between the actin and myosin II dynamics. In myosin II-null cells, actin filaments accumulated in a manner similar to that in wild-type cells (Figure 3C). In the presence of latrunculin A, a depolymerizer of actin filaments, both actin and myosin II filaments disappeared from the cortex independently of wounding, and dividing cells failed to divide, suggesting that the localization of myosin II at the cortex depends on the cortical actin filaments, as described previously [42]. Next, we used jasplakinolide, a permeable actin filament stabilizer. Figure 3D shows the time course of fluorescence intensities when cells expressing GFP-Lifeact were wounded after incubation with jasplakinolide. The duration of actin accumulation significantly increased. The duration of myosin II disappearance also significantly increased when cells expressing GFP-myosin II were wounded after incubation with jasplakinolide (Figure 3E). Figure 3F shows a summary of the initiation, peak, and termination times of actin and myosin II under the above conditions.

These results suggest that actin filaments accumulate at the wound site independently of myosin II, although cortical actin filaments are required for the localization of myosin II at the wound cortex. Myosin II may be dispensable for wound repair based on the results of the PI experiments.

### 3.5. Mechanism for Myosin II Dynamics

Next, we examined the mechanism underlying the disappearance of myosin II upon wounding. The assembly of myosin II filaments in *Dictyostelium* cells is regulated by the phosphorylation of three threonine residues in the tail of heavy chains; their dephosphorylation induces the assembly of filaments, and their phosphorylation induces disassembly [43,44]. The localization of myosin II to the cortex requires the dephosphorylation of heavy chains [43,45]. 3ALA mutant myosin II, carrying alanine residues in place of the phosphorylatable threonine residues, mimics the dephosphorylated state, and therefore localizes only to the cell cortex [43,46]. When myosin II-null cells expressing GFP-3ALA myosin II were wounded, GFP-3ALA myosin II filaments disappeared in a manner similar to that of wild-type myosin II, but their recovery took longer (Figure 4A,C). Therefore, the disappearance of myosin II filaments was not caused by their disassembly. In addition, GFP-3ALA myosin II did not disappear after wounding in the absence of Ca^2+^ (Figure 4C, green), suggesting that the influx of Ca^2+^ induces the myosin II disappearance regardless of its phosphorylation. 

To examine whether the motor activities of myosin II filaments are required for their dynamics, we used myosin-null cells expressing GFP-E476K (motorless) myosin II, which can weakly bind to actin filaments but have no or negligible MgATPase activity [25]. GFP-E476K myosin II filaments also disappeared after wounding, but their recovery took longer (Figure 4B,D). As a complementary experiment, myosin-null cells expressing GFP-wild-type myosin II were wounded in the presence of blebbistatin, an inhibitor of myosin II ATPase. Myosin filaments disappeared in a manner similar to that in the control, but their recovery took longer (Figure 4E). Figure 4F shows a summary of the initiation, peak, and termination times under the above conditions for comparison. 

These results indicate that the disappearance of myosin II filaments from a wound site requires neither regulation via phosphorylation nor motor activities, although their reappearance may require both.

### 3.6. Other Proteins Localizing at the Furrow also Disappeared from the Wound Site

Next, we examined the wound-induced responses of cells expressing GFP-IQGAP-related protein (GAPA), GFP-cortexillin II, GFP-p21-activated protein kinase A (PakA), GFP-myosin heavy chain kinase C (MHCKC), GFP-phosphatase and tensin homolog deleted on chromosome 10 (PTEN G129E), or GFP-dynamin like protein A (dlpA), all of which are known to localize at the cleavage furrow; their knockout mutants showed cytokinesis deficiency [26,27,28,30,36,37,38,47]. Appendix A shows the typical time courses of fluorescence images of cells expressing individual GFP-protein during wound repair. Figure 5A–H show the time courses of fluorescence intensities of individual GFP-protein at the wound site. Interestingly, all these proteins disappeared at the wound site upon wounding, and the black spot expanded. Only GFP-dlpA did not recover for a long time, but the other proteins recovered to resting levels. Since dlpA may be related to clathrin-mediated endocytosis, dividing cells expressing GFP-clathrin light chain (clc) were also wounded. GFP-clc also disappeared and did not recover (Figure 5G). None of the examined proteins disappeared upon wounding in the absence of Ca^2+^ (green in Figure 5A–G), suggesting that Ca^2+^ influx is a trigger for their dynamics. Figure 5H shows a summary of the initiation time of their disappearance, peak, and termination times. The initiation times of GAPA, cortexillin II, dlpA, and clathrin were significantly earlier than that of myosin II, suggesting that these proteins may be located upstream of myosin II dynamics.

Next, mutant cells deficient in individual proteins expressing GFP-myosin II were wounded (Figure 6A–G). Figure 6H shows a summary of the initiation, peak, and termination times. Among the mutants, only the initiation time of GAPA-null cells (4.7 ± 0.3 s, *n* = 25) was significantly earlier than that of wild-type cells (approximately 5.8 s), and this initiation time was similar to that of GFP-myosin II in daughter and interphase cells, suggesting that GAPA was responsible for the delay in the initiation time of myosin II dynamics at the furrow. Further, we examined the effect of W7 on the GFP-GAPA and GFP-cortexillin II dynamics, respectively (Figure 6I,J). In the presence of W7, whereas the GFP-cortexillin II dynamics did not significantly change, the initiation of GFP-GAPA dynamics was significantly delayed and the duration was also prolonged (Figure 6K), suggesting that calmodulin regulates myosin II dynamics via the regulation of GAPA activity.

## 4. Discussion

In this study, we observed the dynamics of wound repair in dividing cells for the first time. We found that actin transiently accumulates and myosin II transiently disappears from the wound site at the furrow cortex in dividing *Dictyostelium* cells. Both actin and myosin II dynamics were dependent on the influx of Ca^2+^ from the external medium. The initiation times of both actin and myosin II dynamics were significantly different from that of Ca^2+^ influx. Intracellular Ca^2+^ levels immediately increased with a peak time of approximately 1.3 s after wounding, whereas actin filaments began to accumulate at approximately 4.1 s and myosin II began to disappear at approximately 5.8 s after wounding. Therefore, Ca^2+^ does not directly induce the actin and myosin II dynamics, but other factors may be involved as intermediates. Actin assembly and myosin II disappearance did not occur only at the wound site, but also spread to a larger area. This spreading rate was approximately 0.33 µm^2^/s. If a diffusible signal spreads from a wound site, the molecular mass is estimated to be approximately 200,000 Da from the diffusion coefficient under consideration of the cytosolic viscosity, which indicates that small molecules such as Ca^2+^ do not directly contribute to the actin and myosin II dynamics. 

The wound responses in dividing cells were different from those in interphase cells. The initiation of actin and myosin II dynamics at the furrow occurred much later than at the cortexes of interphase and daughter cells. The initiation time of myosin II in GAPA-null cells was similar to that in interphase and daughter cells, indicating that GAPA is responsible for the delay in the initiation time at the furrow. Of the four IQGAP (IQ motif containing GTPase activating protein)-related proteins in *Dictyostelium* cells, GAPA (DdIQGAP2) and DGAP1 (DdIQGAP1) localize at the cleavage furrow and are involved in cytokinesis [37,48]. IQGC (DdIQGAP3) mildly affects cytokinesis, although it does not localize at the cleavage furrow [49]. In mammalian cells, IQGAP1 caps the barbed ends of actin filaments; this is inhibited by calmodulin [50]. However, the calmodulin-binding ability of DdIQGAPs has not been verified. GAPA and DGAP1 have been shown to regulate actin dynamics during cytokinesis by interacting with actin-binding proteins such as filamin and cortexillin I [48,51]. Recent studies have shown that GAPA and DGAP1 bind to myosin II and cortexillin I in vitro [52]. In addition, the simultaneous elimination of DGAP1 and GAPA prevents the localization of cortexillins in the cleavage furrow [27]. In the present study, calmodulin regulated both actin and myosin II dynamics. There are many calmodulin-binding proteins in eukaryotic cells, including *Dictyostelium* cells [53]. The target of calmodulin needs to be determined to regulate both actin and myosin II dynamics in the future.

Myosin II localizes at the rear but not at the anterior pseudopods of polarized migrating *Dictyostelium* cells [41]; the mechanism for this has been discussed as follows. Myosin heavy chain kinase A (MHCKA) localizes at the anterior pseudopod, which facilitates the disassembly of myosin II filaments to remove them from the anterior by MHC phosphorylation [45,46]. Monomeric myosin II goes to the rear and is assembled by MHC phosphatase (Figure 7A). Myosin II filaments associated with the cortical actin meshwork exhibit a rapid turnover with a half-life of approximately 7 s (Figure 7B). Their dissociation from the actin cortex is required for their disassembly due to the MHC phosphorylation [24]. Another model for the dissociation of myosin II filaments is that the force generated by myosin motor activities cuts actin filaments, enabling escape from the actin meshwork [42]. However, in this study, we found that neither phosphorylation nor motor activity was involved in the disappearance (dissociation from the cortex) of myosin II filaments upon wounding. Therefore, we propose a novel mechanism for myosin II delocalization from the cortex; the new assembly of actin filaments (presumably, a dense meshwork) excludes myosin II filaments from the cortex (Figure 7C). Gelation actin-binding proteins can exclude myosin II from the dense actin meshwork in vitro [54] and an increase in the thickness of the cortical actin meshwork can exclude myosin II during oocyte maturation [55], which support this model. 

In this study, actin accumulated normally in myosin II-null cells, suggesting that actin dynamics were independent of those of myosin II. In the presence of jasplakinolide, the duration of myosin II disappearance was prolonged. These observations suggest that actin filaments accumulate at the wound site independently of myosin II, and that the disappearance of myosin II depends on actin accumulation. If the Ca^2+^ level increases locally at the anterior region of a migrating cell and induces a newly formed dense actin meshwork in the pseudopod, myosin II filaments can delocalize from the pseudopod in a manner similar to that at the wound site, resulting in their localization at the rear. The exact molecular mechanism remains to be clarified.

All the examined proteins localizing at the furrow disappeared from the wound site depending on the Ca^2+^ influx. Including myosin II, these proteins may be anchored directly or indirectly to cortical actin filaments at the furrow. Cortexillin I and II form a dimeric actin crosslinker and may interact with actin, myosin II, and GAPA. MHCKC contributes to myosin II turnover in the contractile ring [26,56]. PakA is required for the localization of myosin II, although it does not directly phosphorylate it; presumably, it phosphorylates MHCKs [27]. The observation that myosin II transiently disappeared from the wound site in PakA- and MHCKC-null cells, along with the results of 3ALA myosin II, supports the novel mechanism described above. PTEN is a phosphatase that dephosphorylates phosphatidylinositol 3,4,5-triphosphate into phosphatidylinositol 4,5-bisphosphate and localizes at the cleavage furrow, presumably to regulate myosin II accumulation there [30]. Among the five *Dictyostelium* dynamin-like proteins, dlpA and dlpB localize at the cleavage furrow by forming hetero-oligomers [28]. They contribute to stabilizing the actin filaments in the contractile ring. All the proteins that localize at the cleavage furrow may be excluded by the same mechanism as that of myosin II. 

In large cells such as *Xenopus* eggs and *Drosophila* embryos, contractile actomyosin rings appear at the wound region, and their contraction generates the power to close the wound pore. However, in small cells such as yeast, *Dictyostelium*, and mammalian cells, only actin filaments accumulate at the wound site. Presumably, large cells have greater tension on the cell surface and require a larger force to close wound pores. Otherwise, only actin structures associated with many actin-binding proteins close wound pores, or other myosins, such as type I myosin, may substitute for myosin II. Recently, we found that myosin IB accumulated at a wound site (data not shown). Since the present laserportion method is applicable to mammalian cells [33], by comparing with *Dictyostelium* cells, we would like to clarify the general molecular mechanism of wound repair in the future.

## Figures and Tables

**Figure 1 cells-10-01229-f001:**
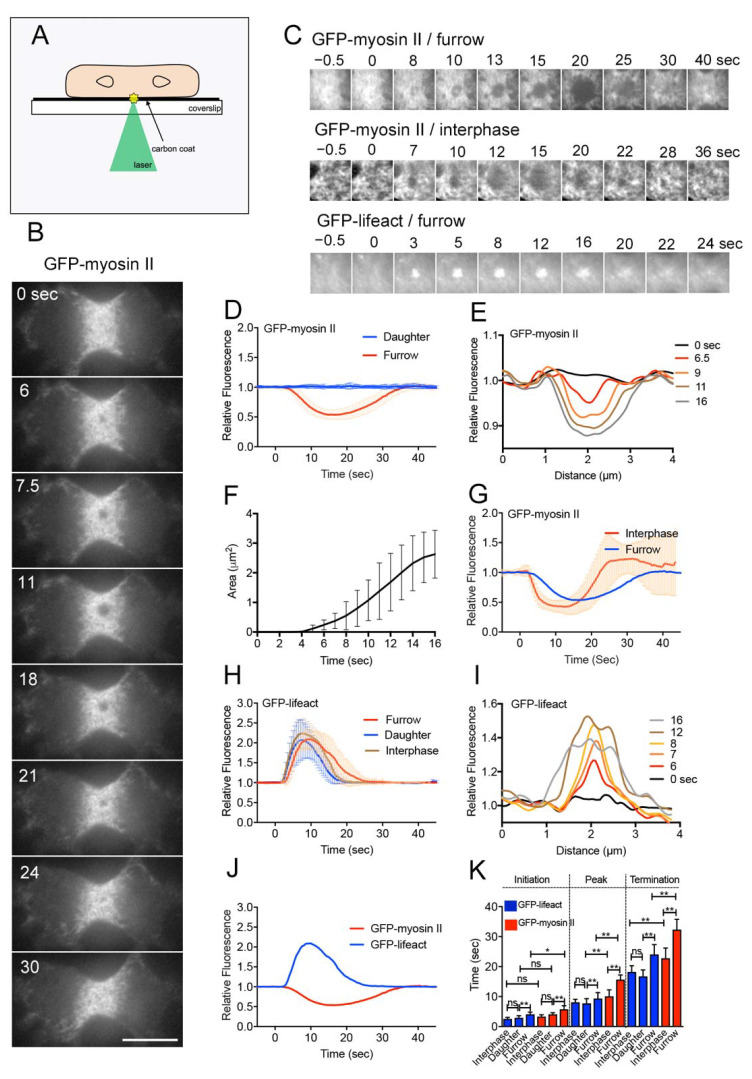
Myosin II filaments transiently disappear from the wound site. (**A**) Schema for laserporation. To make wounds in cell membranes after cells were placed on a carbon-coated coverslip, a laser beam was focused on small local spots beneath single cells under a TIRF microscope. The wound diameter was set as 0.5 μm. (**B**) A typical time course of fluorescence images when a dividing cell expressing GFP-myosin II was wounded at the cell membrane of the cleavage furrow by laserporation. Bar, 10 µm. (**C**) Typical time courses of fluorescence images of GFP-myosin II and GFP-Lifeact in dividing cells, and GFP-myosin II in an interphase cell. Images show only the wound sites. (**D**) Time course of the fluorescence intensities of GFP-myosin II at wound sites in the furrow region and daughter cell cortex (*n* = 25 each). (**E**) A typical time course of myosin dark spot expansion recorded by line-scanning across the wound site. (**F**) Time course of the area of the myosin dark spot (*n* = 25). The graph shows up to 16 s because the boundary of the myosin dark spot became obscure during the recovery. (**G**) Time course of the fluorescence intensities of GFP-myosin II at the wound sites when interphase cells (red, *n* = 25) and the furrow cortex (blue, *n* = 25) were wounded. (**H**) Typical time course of the fluorescence intensities of GFP-Lifeact at wound sites in the furrow and daughter cell cortexes in dividing cells, and the cortex of interphase cells (*n* = 25 each). (**I**) A typical time course of GFP-Lifeact expansion recorded by line-scanning across the wound site. (**J**) Comparison of time courses of the fluorescence intensities of GFP-Lifeact and GFP-myosin II. (**K**) Summary of the initiation, peak, and termination times of actin and myosin II under the above conditions. For GFP-myosin II in daughter cells, only the initiation time was plotted. Data are presented as means ± SD (*n* = 25 each). * *p* ≤ 0.05; ** *p* ≤ 0.0001; ns, not significant, *p* > 0.05.

**Figure 2 cells-10-01229-f002:**
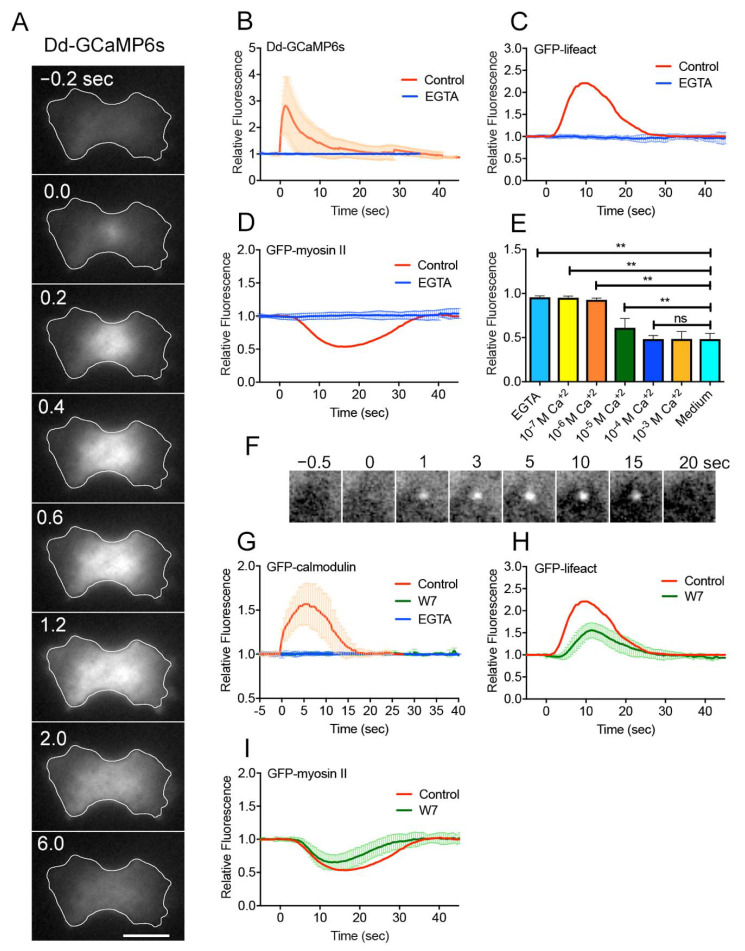
Both myosin II and actin dynamics are regulated by Ca^2+^ and calmodulin. (**A**) A typical sequence of fluorescence images of a dividing cell expressing Dd-GCaMP6s after laserporation. Bar, 10 μm. (**B**) Time course of the fluorescence intensity of cells expressing Dd-GCaMP6s after laserporation in the presence and absence of Ca^2+^(*n* = 25 each). (**C**) Time course of the fluorescence intensity of cells expressing GFP-Lifeact after laserporation in the presence and absence of Ca^2+^. (**D**) Time course of the fluorescence intensity of cells expressing GFP-myosin II after laserporation in the presence and absence of Ca^2+^. (**E**) Peaks of relative fluorescence intensities of GFP-myosin II plotted versus each free Ca^2+^ concentration. Data are presented as means ± SD (*n* = 25 each). ** *p* ≤ 0.0001; ns, not significant, *p* > 0.05. (**F**) A typical sequence of fluorescence images of a dividing cell expressing GFP-calmodulin after laserporation. (**G**) Time course of the fluorescence intensity of cells expressing GFP-calmodulin after laserporation in the presence and absence of Ca^2+^ or in the presence of W7, respectively (*n* = 25 each). (**H**) Time course of the fluorescence intensity of cells expressing GFP-Lifeact after laserporation in the presence and absence of W7 (*n* = 25). (**I**) Time course of the fluorescence intensity of cells expressing GFP-myosin II after laserporation in the presence and absence of W7 (*n* = 25).

**Figure 3 cells-10-01229-f003:**
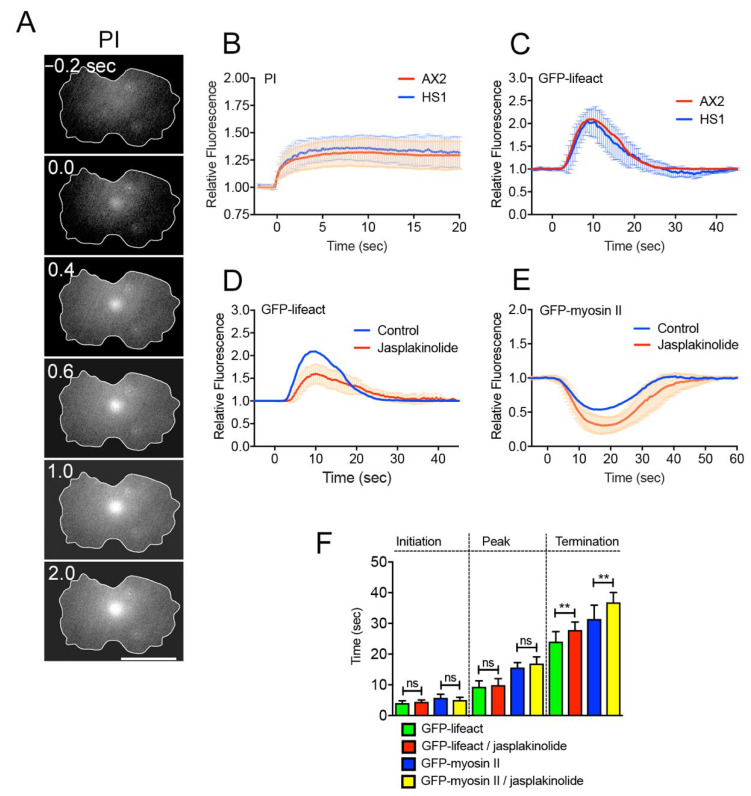
Role of myosin II in wound repair. (**A**) A typical sequence of fluorescence images of propidium iodide (PI) influx after laserporation. Bar, 10 μm. (**B**) Time course of PI influx when the furrows of wild-type (AX2) and myosin II-null (HS1) cells were wounded. The fluorescence intensities were measured inside the outlines of the cells (n = 25 each). (**C**) Time course of the fluorescence intensities of wound sites in AX2 and HS1 cells expressing GFP-Lifeact (n = 25 each). (**D**) Time course of the fluorescence intensities of wound sites in AX2 cells expressing GFP-Lifeact in the presence and absence of jasplakinolide. (**E**) Time course of the fluorescence intensities of wound sites in AX2 cells expressing GFP-myosin II in the presence and absence of jasplakinolide. (**F**) Summary of the initiation, peak, and termination times of actin and myosin II in the presence and absence of jasplakinolide. Data are presented as means ± SD (*n* = 25 each). ** *p* ≤ 0.0001; ns, not significant, *p* > 0.05.

**Figure 4 cells-10-01229-f004:**
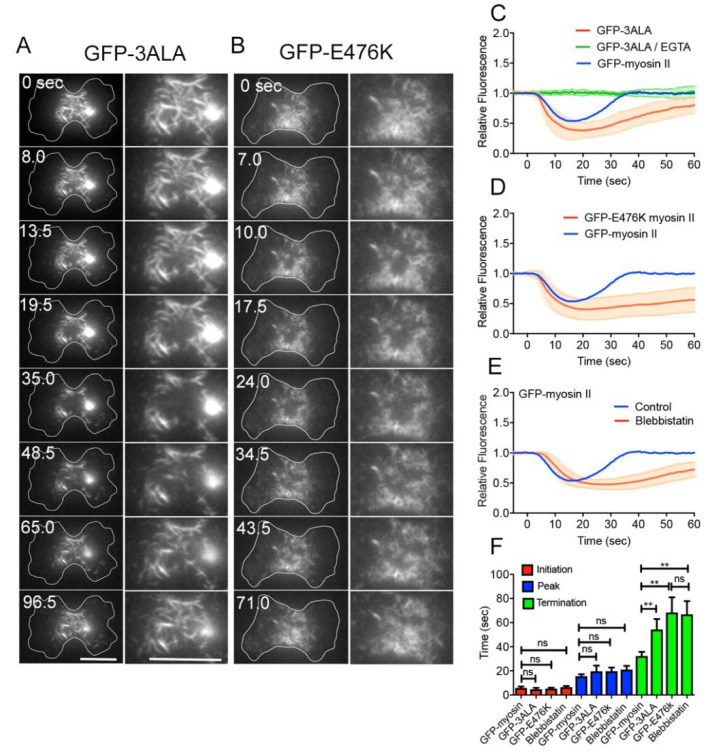
Mechanism for myosin II dynamics. (**A**,**B**) Typical sequences of fluorescence images of GFP-3ALA and GFP-E476K myosin II after laserporation. The individual right columns show only furrow regions. Bars, 10 μm. (**C**) Time course of the fluorescence intensities of GFP-wild-type and GFP-3ALA myosin II after laserporation. The time course of GFP-3ALA myosin II in the absence of Ca^2+^ is also plotted. (**D**) Time course of the fluorescence intensities of GFP-wild-type and GFP-E476K myosin II after laserporation. The graph of GFP-E476K myosin II was plotted until 60 s, but this mutant myosin II finally recovered. (**E**) Time course of the fluorescence intensities of GFP-myosin II in the presence and absence of blebbistatin. (**F**) Summary of the initiation, peak, and termination times under the above conditions. Data are presented as means ± SD (*n* = 25 each). ** *p* ≤ 0.0001; ns, not significant, *p* > 0.05.

**Figure 5 cells-10-01229-f005:**
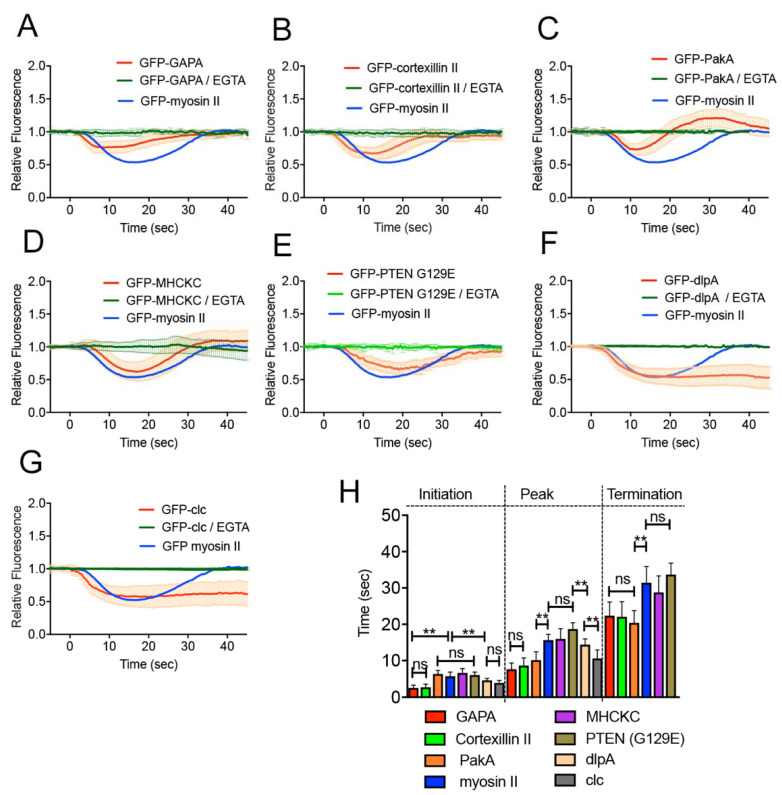
Other proteins localizing at the furrow also disappeared from the wound site. (**A**–**G**) Time courses of the fluorescence intensities of (**A**) GFP-GAPA, (**B**) GFP-cortexillin II, (**C**) GFP-PakA, (**D**) GFP-MHCKC, (**E**) GFP-PTEN (G129E), (**F**) GFP-dlpA, and (**G**) GFP-clathrin light chain after laserporation in the presence and absence of Ca^2+^. As a comparison, the time courses of GFP-myosin II are also plotted (blue). None of the GFP-proteins showed any wound response in the absence of Ca^2+^ (green). We used GFP-PTEN (G129E) for the localization of PTEN, in which the glycine residue at position 129 was replaced with glutamate, and which previously showed negligible phosphatase activity and localization similar to that of wild-type PTEN [48]. (**H**) Summary of the initiation, peak, and termination times of individual protein dynamics. Data are presented as means ± SD (*n* = 25 each). ** *p* ≤ 0.0001; ns, not significant, *p* > 0.05.

**Figure 6 cells-10-01229-f006:**
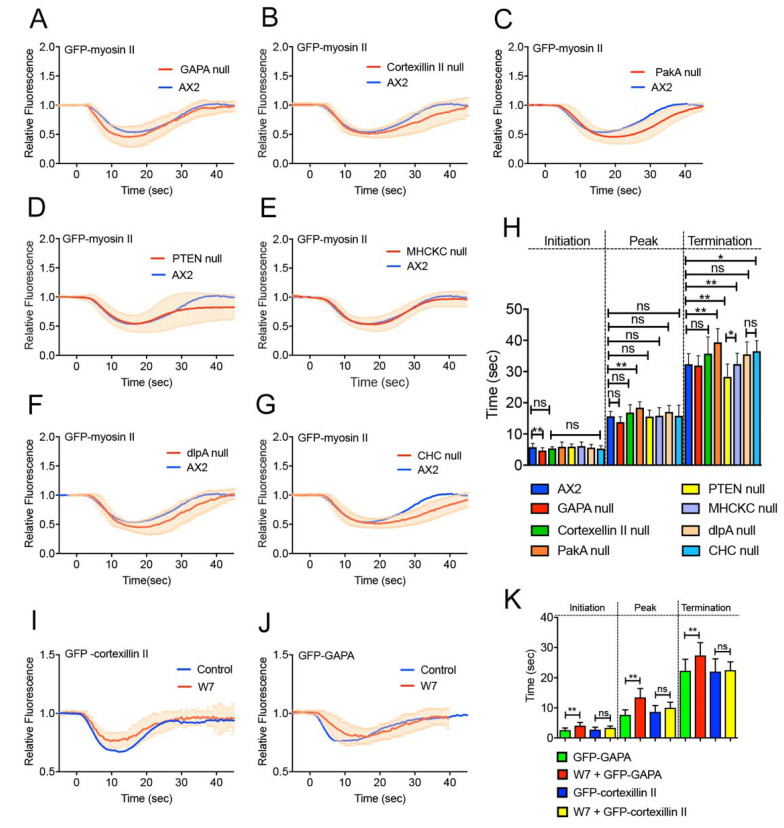
Myosin II dynamics in knockout mutant cells. Time courses of the fluorescence intensities of individual mutant cells expressing GFP-myosin II. (**A**) GAPA-null, (**B**) cortexillin II-null, (**C**) PakA-null, (**D**) PTEN-null, (**E**) MHCKC-null, (**F**) dlpA-null, and (**G**) clathrin heavy chain-null cells. (**H**) Summary of the initiation, peak, and termination times of GFP-myosin II in individual mutant cells. (**I**,**J**) Time course of the fluorescence intensities of cells expressing GFP-GAPA (**I**) and GFP-cortexillin II (**J**) in the presence and absence of W7, respectively. (**K**) Summary of the initiation, peak, and termination times of GFP-GAPA and GFP-cortexillin II in the presence and absence of W7. * *p* ≤ 0.001; ** *p* ≤ 0.0001; ns, not significant, *p* > 0.05.

**Figure 7 cells-10-01229-f007:**
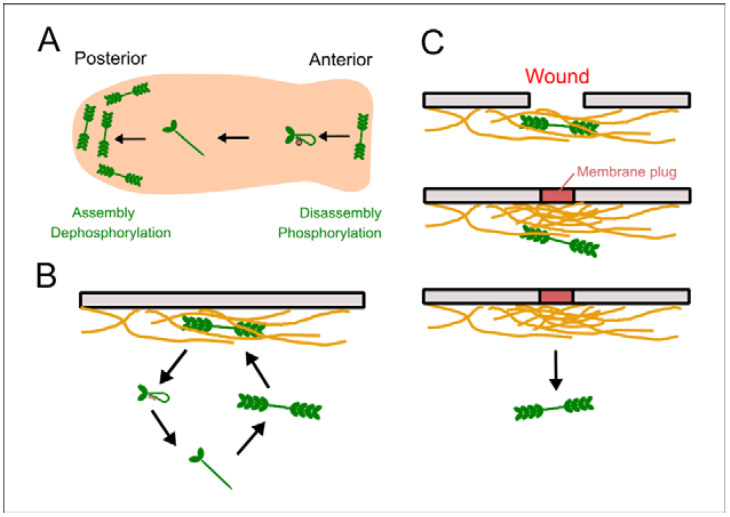
A novel model for the release of myosin II filaments from the cortex. (**A**) Myosin II filaments (green) localize at the rear of migrating cells. Myosin II filaments disassemble into monomers via phosphorylation by MHC kinase A, which localizes at the anterior. The monomers go to the rear and are assembled into filaments by MHC phosphatase. (**B**) Myosin II filaments undergo rapid turnover, repeating their association and dissociation with the actin cortex. When myosin II filaments associated with the cortical actin filaments (orange) are phosphorylated, they disassemble and dissociate from the cortex. The phosphorylated myosin II is dephosphorylated by MHC phosphatase, reassembles into filaments, and binds to the cortical actin again. (**C**) During wound repair, actin filaments are newly assembled by actin-crosslinking proteins and exclude myosin filaments without disassembly.

## Data Availability

All relevant data are available from the authors on reasonable request.

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
