# Peer review of "Dynamics of Myosin II Filaments during Wound Repair in Dividing Cells"

_cells, 2021, doi:10.3390/cells10051229_

Round 1

Reviewer 1 Report

This work has clarified the mechanisms of myosin dynamics during wound repair.  The paper was well-written and well-ordered all the parts. The novelty of the work is high. Thus, I recommend this manuscript for publication.

Author Response

We would like to greatly appreciate for the reviewers helpful reviewing.

Reviewer 2 Report

“Dynamics of Myosin II Filaments during Wound Repair in Dividing Cells”

Tanvir et al. described a novel method for studying wound repair in dividing Dictyostelium cells. In this model, Dictyostelium cells were transformed with plasmids for fluorescent tracking and wounded via laserporation. Specifically, the mechanism of action in wound repair of myosin II, actin, Dd-GCaMP6s, calmodulin, GAPA, cortexillin II, PakA, MHCKC, PTEN G129E, and dlpA were studied. Additionally, the presence of Ca2+, depolymerization of actin filaments, phosphorylation of myosin II, were observed in relation to wound repair. The authors conclude that injury located at the furrow cortex will result in transient actin accumulation and myosin II disappearance, which is dependent on the influx of Ca2+ into the cell. GAPA is responsible for the delayed actin and myosin II dynamics post injury. Calmodulin contributes to the accumulation of actin at the wound site. Phosphorylation and motor activity did not have a direct effect in the disappearance of myosin II after injury. Based on their data, the authors suggest that actin filaments will assemble at the site of injury on the cortex without myosin II filaments present.

This research sheds light on the dynamics of actin and myosin II in dividing cells post-injury. The implication of this research can lead to a better understanding of how defects in cell membrane repair occur and how to prevent such defects that lead to harmful outcomes.

Major Weaknesses

  1. While this is a novel method for studying cellular repair in dividing cells, the authors used only amoeba in their study. Because the authors discuss how cell membrane defects may lead to a variety of pathologies, the use of mammalian cells would help strengthen their data. At the very least, the authors should include significant discussion about this mechanism of action in mammalian cells.

Minor Weaknesses

  1. A more detailed description of the image processing would be appreciated. How were the fluorescence intensities normalized: using the entire image/cell or just at the site of laserporation? Was a macro created to measure the dark spot expansion for the line scanning across the wound site, or was this done manually? If performed manually, how was the consistency maintained?
  2. In figure 1F, the full-time course of injury and recovery should be visualized. The authors claim that myosin II repopulates the site of laserporation at 30 seconds post injury. Showing the area of the myosin dark spot decrease from 16 seconds to 30 seconds would strengthen this claim.
  3. The color scheme in figures 2B-2D should be consistent, such that the same color should be used for control lines throughout the three plots.

Author Response

First of all, we would like to greatly appreciate for the reviewer’s helpful reviewing.

C1: While this is a novel method for studying cellular repair in dividing cells, the authors used only amoeba in their study. Because the authors discuss how cell membrane defects may lead to a variety of pathologies, the use of mammalian cells would help strengthen their data. At the very least, the authors should include significant discussion about this mechanism of action in mammalian cells.

A1: We have already shown that the present laserportion method is applicable to mammalian cells (Yumura, 2016,Scientific Reports 6: 22055). We would like to examine mammalian cells by using the present method in future.

The last line in Discussion: Since the present laserportion method is applicable to mammalian cells [33], by comparing with Dictyostelium cells, we would like to clarify the general molecular mechanism of wound repair in future.

C2: A more detailed description of the image processing would be appreciated. How were the fluorescence intensities normalized: using the entire image/cell or just at the site of laserporation? Was a macro created to measure the dark spot expansion for the line scanning across the wound site, or was this done manually? If performed manually, how was the consistency maintained?

A2: We would like to add the measurement of fluorescence intensities in Methods section as follows.

Line 115: The time courses of the fluorescence intensities within the circle (3 µm in diameter) including the wound site were examined using Image J (http://rsbweb.nih.gov/ij). The fluorescence intensities were normalized by setting the value before wounding to 1 after subtracting the background.

C3: In figure 1F, the full-time course of injury and recovery should be visualized. The authors claim that myosin II repopulates the site of laserporation at 30 seconds post injury. Showing the area of the myosin dark spot decrease from 16 seconds to 30 seconds would strengthen this claim.

A3: The full-time course of injury and recovery are shown in Fig1B and C. As described in the text, the myosin filaments did not recover from the wound boundary. Therefore, the boundary of the myosin black spot is too obscure for us to examine the exact area. The purpose of Figure 1F is just to show the speed of the expansion. We would like to add a following sentence to avoid the confusion.

Line 202: The graph shows up to 16 s because the boundary of the myosin dark spot became obscure during the recovery.

C4: The color scheme in figures 2B-2D should be consistent, such that the same color should be used for control lines throughout the three plots.

A4: We would like to change the color according to the reviewer’s suggestion.

Several grammatical mistakes were fixed as shown in red in the text.